# Generating Self-Shaped 2D Aluminum Oxide Nanopowders

**DOI:** 10.3390/nano12172955

**Published:** 2022-08-26

**Authors:** Meng-Ying Lee, Fu-Su Yen, Hsing-I Hsiang

**Affiliations:** Department of Resources Engineering, National Cheng Kung University, Tainan 70101, Taiwan

**Keywords:** 2D powders, top-down, layered material, thermal-assisted exfoliation, transition alumina, cleavage, pseudomorph

## Abstract

The thermal-assisted exfoliation phenomena of boehmite particles under moderate heating rates were examined. The exfoliation that generated flakes of 5–6 nm in thickness can be achieved because of the perfect cleavage on the boehmite particles that are stripped when thermal treatments bring about dehydration and γ-Al_2_O_3_ formation in sequential phase transformation of boehmite. Examinations of the exfoliation effects were carried out on calcined boehmite single crystal particles, which were about 500 nm in diameter, and obtained at three heating rates 0.5, 1.0, and 2.0 °C/min with the heating schedules. The TEM techniques, BET-N_2_ measurements, XRD-Scherrer equation, and AFM images were employed in the examination. That the BET values increased as increasing of exfoliated flakes reflected two stages of exfoliation. In the beginning stage, during which the BET values were <40 m^2^/g, the exfoliation resulted from the stress produced by dehydration. In the second stage, the increased rate of surface area was due to the additional force, which originated from the γ-Al_2_O_3_ formation. Exfoliation occurred on the cleavage planes {010}, the side pinacoid of the boehmite particle. The generation of flakes resulted in the thinning of boehmite particles. Some of the flakes preserved the external form of boehmite crystals. From the surface energy evaluations of boehmite and γ-Al_2_O_3_, it can be inferred that exfoliation is a natural way of thermal treatment.

## 1. Introduction

Ultrathin two-dimensional (2D) nanomaterials, with their unique and indispensable high specific surface area [1] and relatively large lateral size, offer some novel physical, chemical, and electronic properties due to their electron confinement in 2D [2]. A wide range of applications was developed, such as electronics/optoelectronics [3,4], catalysis [5,6], and thermal wear coatings [7,8]. The preparation methods of ultrathin 2D nanomaterials can be categorized into top-down and bottom-up two approaches. Top-down approaches were applied to those layered compounds, such as graphene [9,10,11,12], h-BN [13,14,15,16], and MoS_2_ [17,18,19]. Most of the compounds display good cleavage planes that can be exfoliated using Scotch tape [20] or other microchemical cleavage techniques [21,22,23], including electrochemical exfoliation [18,24,25,26] and thermal treatments [10,11,27,28] depending on voltage-driven or released gas between interlayers or sheets followed by expansion and finally exfoliation. It is non-destructive but normally has high technical requirements. Bottom-up approaches have been used to produce inorganic 2D nanomaterials [29], such as transition metal oxides (e.g., TiO_2_, ZnO, Co_3_O_4_) [30] and metal chalcogenides (e.g., PbS, CuS, CuSe, SnSe) [31], etc. CVD techniques [19,32,33] and other wet-chemical synthesis methods [34,35,36] were adopted. However, these methods are complicated in technique and relatively high cost.

The concept offered in this study is the utilization of autogenous forces originating from thermal treatment so as to break the cleavage planes to produce flakes. The raw material used in this study is an aluminum hydroxide mineral, boehmite, which is a lamellar-structured mineral with the prominent perfect cleavage {010}_b_(side pinacoid) on the particles [37]. Boehmite is transformed to γ-Al_2_O_3_ when the temperature reaches 450 °C and boehmite undergoes a topotactic transformation to another layered oxide, γ-Al_2_O_3_ [38]. The side pinacoid {010}_b_ of boehmite and the tetragonal prism {110}_γ_ of γ-Al_2_O_3_ are the most relevant crystallographic orientations [39,40,41,42], representing the same and preferentially exposed surface on the particle.

The phase transformation to γ-Al_2_O_3_ provokes the release of structural water and an increase in molar density from 3.07 to 3.71 g/cm^3^. The former generates water vapor pressure between the cleavage planes [43,44]. The latter results in dimension difference between boehmite and the new-formed γ-Al_2_O_3_ [45,46], which may trigger displacement on either side of the cleavage planes [47].

Table 1 lists the orientation relationship between boehmite and γ-Al_2_O_3_ and schematically illustrates the views of cleavage surfaces {010}_b_ and {110}_γ_ to emphasize that these two cleavage planes are equivalent. Diffraction data from the powder diffraction file [48] (In text: PDF 21-1307 and PDF 29-0063) were used to simulate the characteristic morphology. If boehmite is converted to γ-Al_2_O_3_, the axial lengths of a- and c- of boehmite vary by +7.1% and −2.3%, respectively. The interfacial angles constituting the cleavage planes {010}_b_ and {110}_γ_ are 75.5° (boehmite{101}Ʌ{101}) and 70.5° (γ-Al_2_O_3_ {111}Ʌ{111}). The dimension difference induces a displacement force on either side of the cleavage planes, resulting in the occurrence of exfoliation.

Bearing this in mind, it is of great importance to offer the concept from this topic that the hydroxide minerals with a laminated structure have the basis for the generation of flake powders. The process does not have complex steps and is environmentally friendly, and the operation method can be further improved to make flake powder with a high specific surface area. In addition to extending the current understanding of boehmite, the results will be applicable to other similar and related systems such as gibbsite [49,50], γ-FeOOH [51], Co(OH)_2_ [26,52], and Cd(OH)_2_ [53,54].

In this paper, the exfoliation of boehmite under moderate thermal treatment was investigated. Particle size (especially thickness), morphology, and the mechanism of flake formation were also evaluated. The examination was carried out on calcined boehmite single crystal particles with a diameter of about 500 nm at three heating rates of 0.5, 1.0, and 2.0 °C/min at predetermined temperatures using TEM techniques. The thickness of the flakes was estimated by BET-N_2_ measurements and XRD-Scherrer equation and finally compared with AFM images.

## 2. Experimental Procedure

### 2.1. Material

High purity (>99.9%) boehmite powders, γ-AlO(OH) (Shandong Gemsung Crystal Technology Co., Ltd. Zibo City, China) were used in this study. The as-received powders were de-agglomerated using ball milling in deionized water (solid content of ~30 wt%, pH = 4.0 adjusted by 2N HNO_3_) using 3 mm diameter alumina balls. The milled slurry was passed through a 400-mesh standard sieve (<38 μm). After drying in a microwave oven, the dried powder was thermally treated at predetermined heating schedules to obtain test samples. The basic properties and characteristics of the starting boehmite powders are shown in Table 2 and Figure 1 and Figure 2.

By heating the starting boehmite powders at the heating rates of 0.5 °C/min (450 °C~510 °C), 1.0 °C/min (460 °C~520 °C), and 2.0 °C/min (470 °C~530 °C). The calcined samples (2–3 g each) were collected every 5~10 °C to examine exfoliation effects and the transformation process. The predetermined temperature ranges shown in parentheses were obtained using the DTA technique under heating rates of 0.5, 1.0, and 2.0 °C/min, respectively. When the set temperature was reached, the sample was removed directly from the furnace and allowed to cool to room temperature (cooling rate > 150 °C/min). Please refer to Appendix A: Figure A1 for the phase transformation under each condition.

### 2.2. Characterization

The differential thermal analysis (DTA, Labsys evo, Setaram, Lyon, France) technique under the above three heating rates was employed to determine the endothermic temperature ranges induced by dehydration of boehmite and the temperature of boehmite transition to γ-Al_2_O_3_. The weight losses (%) of samples before and after the thermal treatments were recorded ((W_before_ − W_after_) × 100%/W_before_). The fractions of boehmite and γ-Al_2_O_3_ in the calcined samples were determined by quantitative XRD (Rigaku MiniFlex, Rigaku Corp, Tokyo, Japan) method using Ni-filtered Cu*K*α radiation with CaF_2_ (10 wt%) as the internal standard. The integrated intensities of boehmite: (010)_b_ (2θ = 14.45°), γ-Al_2_O_3_:(222)_γ_ (2θ = 39.36°), and CaF_2_:(111) (2θ = 28.27°) were measured. Their ratios were then compared with the boehmite/γ-Al_2_O_3_-CaF_2_ calibration curves. The variation of specific surface area values [55] of the powder system was measured by BET (Micromeritics Gemini 2390, Norcross, GA, USA) to identify the exfoliation progress. Morphological investigations including cross-sectional sizes (Feret’s diameter [56]) distribution of the calcined samples in each stage were performed by TEM (FEG-TEM, Tecnai G2 F20, FEI, Hillsboro, OR, USA) and AFM (JPK Nanowizard AFM, Axiovert 200, Berlin, Germany) analysis.

In this study, measurements of the thickness of the obtained powders after thermal treatments were carried out in three ways:Mean thicknesses of the calcined boehmite/γ-Al_2_O_3_ flakes

The mean thicknesses of calcined boehmite/γ-Al_2_O_3_ flakes were derived by measuring the BET-specific surface area values and expressed in terms of *t_BET_*. It was assumed that the calcined powder is composed of flake particles with a mean cross-sectional diameter of *D* (in this case, about 450 nm). The thickness *t_BET_* was then calculated by (Equation (1)) (Appendix B, Figure A2).
(1)$$t_{BET}\left(\mathrm{nm}\right)=\frac{2D}{\rho \times D\times BET\;S.S.A.\;\times 10^{-3}-4}$$
where *D*: Diameter in average of the cross-sectional area (450 nm)

*ρ*: Density of boehmite/γ-Al_2_O_3_ (g/cm^3^)

and *S.S.A.* (Specific surface area): BET specific surface area value (m^2^/g)

2.Thickness of boehmite and γ-Al_2_O_3_ flakes

The thickness of boehmite and γ-Al_2_O_3_ flakes were calculated by the XRD-Scherrer equation [57],
(2)$$D_{\mathrm{hkl}}=\frac{K\lambda }{B\cos\theta }$$
where *D*_hkl_ is the crystallite size in the direction perpendicular to the lattice planes, *K* is a numerical factor and is usually taken as about 0.89, λ is the wavelength of the X-rays (Cu Kα1 radiation ((1.540562 Å)), *B* is the Full width at half maximum (FWHM) in radians, and θ is the Bragg angle. The reflection peaks of boehmite: (020)_b_(2θ = 14.5°) and γ-Al_2_O_3_:(440)_γ_(2θ = 66.7°) were used in the cleavage directions, and the 2θ of 13.5°–15.5° for boehmite and 64°–70° for γ-Al_2_O_3_ with a scanning rate of 0.5°/min were applied. The peak widths of the instrument were calibrated with well-crystallized silicon powder. Data calculations were performed with the assistance of software, XRD pattern processing, and identification, Jade for Windows, Version 5.0. The obtained thickness of boehmite and γ-Al_2_O_3_ flakes are expressed in terms of *t*_B{010}_ and *t*_γ{110}_.

3.AFM

Samples were imaged by AFM in intermittent contact mode. The probe was a Tap 150Al-G silicon probe.

## 3. Results and Discussion

### 3.1. The Effects of Thermal Treatment on Boehmite

The thermal treatment on boehmite brings about its dehydration and the phase transformation into γ-Al_2_O_3_. As the heating proceeded, the weight loss was accompanied by an increase in the specific surface area (BET-N_2_), which is related to the exfoliation of the calcined boehmite. Figure 3 illustrates the BET values (Figure 3a) and mineral phase changes (Figure 3b) in relation to the weight loss for samples prepared at three heating rates. At the beginning of the thermal treatment, the dehydration induced a 2% weight loss of physically absorbed water; it was then followed by a 13% weight loss of the structural water. Further heating at temperatures above 600 °C resulted in a weight loss of about 2%.

In Figure 3a, the relationship of specific surface area with the weight loss showed a turning point at 40 m^2^/g or about 7% of the weight loss, and then the slope became steeper until the specific surface area reached 100 m^2^/g. Figure 3b shows the disappearance of boehmite and the formation of γ-Al_2_O_3_ in relation to weight loss. It can be seen that the boehmite content began to decrease when the weight loss reached 2%. However, the formation of γ-Al_2_O_3_did not start until the weight loss reached 7% or the BET value reached 40 m^2^/g, which is comparable to the turning point shown in Figure 3a.

If the increase in BET value is related to the exfoliation of boehmite particles, from the BET value turning point in Figure 3a,b, it is assumed that the surface area increase can be divided into two stages, each of which was dominated by different operating forces. In the beginning, the operating force was mainly contributed by dehydration until the BET value reached 40 m^2^/g. The increased rate suggested that in the second stage, an additional force was derived from the dimension difference of the cleavage planes due to the presence of γ-Al_2_O_3_. The exfoliation was first caused by the water vapor pressure generated by the release of structural water of boehmite and was completely controlled solely by water vapor. As the weight loss increased to 7% or the BET value reached 40 m^2^/g, the dimension difference, which occurred from the phase change to γ-Al_2_O_3_, added another operating force to the process.

The quantitative phase analysis (Figure 3b) showed that the disappearance of boehmite and the formation of γ-Al_2_O_3_ did not carry on synchronously during the phase transformation. It can be seen that the γ-Al_2_O_3_ phase started to appear after nearly 30% of the boehmite disappeared. It is worthwhile to note that this 30% of the boehmite actually transformed to boehmite-derivative before the appearance of γ-Al_2_O_3_, which is characterized by the released or partially released structural water. The boehmite-derivative may be a transition state between structurally collapsed boehmite and structurally established γ-Al_2_O_3_ [58]. The morphology of the {010}_b_ cleavage plane inherited from boehmite is easily found by TEM (Figure 4). In this study, this derivative is tentatively designated as meta-γ. The meta-γ content decreased with the increase of thermal treatment temperature. When boehmite disappeared completely, the meta-γ content was also close to zero, and γ-Al_2_O_3_ content was close to 100%.

### 3.2. The Characteristics of Flakes

#### 3.2.1. Four Types of Flakes Particles

For the calcined samples, four types of lamellar particles can be observed. Figure 4a shows the morphology of boehmite, which was classified as a powder system with a BET value of less than 40 m^2^/g. Figure 4b shows the morphology of meta-γ, which verified that the ratio of d-values (obtained by SAED) in both directions of the cleavage planes lies between boehmite and γ-Al_2_O_3_ (Table A1). The highest ratio, about 30%, can be found in samples with BET values of around 40 m^2^/g. Figure 4) shows γ-Al_2_O_3_ particles existed in a pseudomorph form after boehmite. This is usually found in calcined samples with BET values from 40 to 80 m^2^/g. Figure 4d shows the γ-Al_2_O_3_ flakes per se and the complete structure and morphology. The γ-Al_2_O_3_ flakes are mainly present in calcined samples with BET values above 80 m^2^/g (Table A2).

#### 3.2.2. The Thickness Evolution of Particles

Figure 4 shows the average thickness *t*_BET_, which was calculated by Equation (1), as well as the BET values of the calcined boehmite. The same figure also shows the thickness of boehmite, *t*_B{010}_, γ-Al_2_O_3_, *t*_γ{110}_, and meta-γ, *t_meta_*_(BET)_ in the calcined samples at the corresponding BET values. The thickness *t*_B{010}_ and *t*_γ{110}_ were calculated by the XRD-Scherrer equation (Equation (2)), and *t_meta_*_(BET)_ was calculated by Equation (1) based on the phase content ratios and the thickness of coexisting boehmite and γ-Al_2_O_3_. According to the results, *t_meta_*_(BET)_ has an average value of ~6 nm, which is comparable to the value of *t*_γ{110}_. It is worthwhile to mention that the thickness of γ-Al_2_O_3_ was from 5 to 6 nm after its appearance (estimated by Equation (1) or Equation (2)). Therefore, the thicknesses of γ-Al_2_O_3_ and meta-γ can be considered equivalent.

From the thickness data shown in Figure 5, it can be deduced that the thinning process of boehmite started from the initial *t*_B{010}_ values of 55–49 nm before the appearance of γ-Al_2_O_3_. After that, the thinning process accelerated until *t*_B{010}_ reached the final minimum value of 22nm. While both γ-Al_2_O_3_ and meta-γ as exfoliated products exhibited almost the same thickness, 5–6 nm, and remained essentially constant throughout the phase transition. Further examination using the AFM technique also confirmed the above results (see Figure 6).

It is clear that thermal treatments on boehmite can produce exfoliation effects on boehmite particles and bring about an increase in specific surface area (BET values). Table 3 demonstrates the changes of calcined samples in mineral phase content, particle shape ratio, and thickness of flakes for samples of BET values of 40, 70, 90, and 100 m^2^/g. The exfoliation of boehmite particles, depicted by the BET values, continued as the thermal treatment proceeded. The BET values are contributed by residual boehmite particles and newly formed γ-Al_2_O_3_ and meta-γ flakes. The boehmite particles gradually became thinner due to exfoliation, and the γ-Al_2_O_3_ and meta-γ flakes remained at a constant thickness of 5–6 nm. When boehmite was completely transformed, flake powders of γ-Al_2_O_3_ with a thickness of 5–6 nm were obtained.

Figure 7 demonstrates that at a moderate heating rate, the thermal treatment to produce exfoliation may begin from small particles. A rough statistical sampling of the number of flakes with different cross-sectional sizes was carried out on TEM images of samples with BET values of 40, 70, and 90 m^2^/g. It reveals that in the early stage (40 m^2^/g), exfoliation mainly occurred on small-sized particles, resulting in a high percentage of small flakes. As the BET values increased, i.e., in the later stage, relatively large flakes appeared.

#### 3.2.3. Boehmite Relics in γ-Al_2_O_3_ Flakes

Because thermal-assisted exfoliation occurs on the cleavage plane {010}_b_ of boehmite, the generated γ-Al_2_O_3_ flakes should possess the topotactic nature of boehmite. These γ-Al_2_O_3_ flakes apparently inherited the morphology or particle shape of boehmite, forming a pseudomorph after boehmite (Figure 4c). Apparently, at higher temperatures, some of the flakes transformed into γ-phase morphology (Figure 4d). Samples with high BET values have the possibility to contain more γ-Al_2_O_3_ flakes with γ-Al_2_O_3_ morphology. Detailed examination reveals that the flakes in pseudomorphism decreased from 48 to 42% as the BET values increased from 70 to 90 m^2^/g (Table 3). When the BET value approached 100 m^2^/g, about 35% of γ-Al_2_O_3_ in the powder system showed pseudomorph after boehmite (Table 3). It is known that the thermal treatment of the boehmite particles causes the release of structural water, which will initiate the exfoliation of boehmite particles. The exfoliation occurs only on the cleavage plane {010}_b_, i.e., the planes of symmetry at the ends of the boehmite particles in the b-axis direction.

The generated γ-Al_2_O_3_ flakes showed a thickness of 5–6 nm with a slight shrink in size and preserved the appearance of boehmite (external form). This is significant because it causes the thinning of boehmite particles. Higher processing temperatures promote the appearance of γ-Al_2_O_3_ flakes. In addition, the increase in BET value and γ-Al_2_O_3_ content is accompanied by a decrease in meta-γ, indicating the energy input is crucial for the phase transformation and favors the formation of large-sized flakes.

### 3.3. Exfoliation Model

#### 3.3.1. Two-Staged Procedures

The exfoliation process can be divided into two stages according to where the dominating operating forces originated from. The first stage occurred before the BET value reached 40 m^2^/g. In the early stage of the phase transformation, the operating force came from the dehydration caused by the thermal treatments. The water vapor ejected from boehmite particles during dehydration broke the cleavage planes of the boehmite. This force dominated in the first-stage exfoliation. The second stage began when BET value was greater than 40 m^2^/g. When the new phase γ-Al_2_O_3_ appeared, the strain due to shrinkage in dimension arising on the interface (cleavage plane) between the boehmite and γ-Al_2_O_3_ became an additional force. Therefore, in the second stage, both shrinkage strain and water vapor pressure coexisted, leading to the exfoliation.

#### 3.3.2. Exfoliation Details

##### Stage One

As dehydration continued, boehmite vanished, and its content dropped to ~70%. The meta-γ with a thickness of 6 nm was formed. The powder system consisted of boehmite and the newly formed meta-γ flakes, which are stripped from the cleavage planes {010}_b_, the side pinacoid of the boehmite particle. In this stage, the average thickness of the boehmite particles was getting thinner, from 55 nm to 49 nm. When the BET value reached 40 m^2^/g, the meta-γ content was about 30%, and the γ-Al_2_O_3_ flakes, which had been converted from meta-γ flakes, appeared. This is the end of the first stage of exfoliation.

##### Stage Two

In the second stage, γ-Al_2_O_3_ appeared, and the BET value exceeded 40 m^2^/g. As the temperature further increased, the content of boehmite decreased and finally became null. The higher temperature favored the formation of meta-γ and its conversion to γ-Al_2_O_3_. The boehmite content continued to be decreased, and the average thickness decreased from 49 to 22 nm. The meta-γ flakes, as formed in the first stage, exfoliated from the side pinacoid of the boehmite particle and left the relics of boehmite. In addition, the formation and conversion rate of meta-γ accelerated, resulting in a high percentage of γ-Al_2_O_3_ formation and nearly absence of meta-γ. Finally, 2D alumina flake powders with a mean thickness of 5–6 nm were obtained while the BET value reached ~100 m^2^/g. Figure 8 illustrates the interpretation of the exfoliation model that occurred in this study.

#### 3.3.3. Thermodynamics of the Flake Formation

Figure 4 demonstrates the TEM micrographs of the four types of flake particles observed in this study. It is interesting to note that all particles show similar morphology and obviously are the derivatives of boehmite {010}_b_ cleavage planes. The calculated and measured thickness of the meta-γ and γ-Al_2_O_3_ flakes were 5–6 nm. Meanwhile, the final thickness of boehmite was 22 nm (calculated from BET value ~90 m^2^/g), which is close to 3 or 4 times 6nm. The calculations in Table 4 show that the γ-Al_2_O_3_ particles with D_50_ of 450 nm and thickness of 5.7 nm and the boehmite particles with D_50_ of 450 nm and thickness of 22 nm have similar surface equivalent energies per unit flake (boehmite: 2.38 × 10^−10^ mJ/flake, γ-Al_2_O_3_: 5.05 × 10^−10^ mJ/flake). This suggests that 2D powders can be generated in a natural way, cleaving or exfoliating during thermal treatment.

## 4. Conclusions

In summary, applying the thermal treatment to boehmite particles can produce exfoliation effects on it. Since boehmite exhibits perfect cleavage on the side pinacoid, {010}, by which boehmite has the potential to be stripped into flakes or exfoliation. In the thermal treatment, boehmite is transformed to γ-Al_2_O_3_. The complete transformation process includes the dehydration and γ-Al_2_O_3_ formation, in which the water vapor pressure and shrink strain were induced, respectively. From the surface energy evaluations of boehmite and γ-Al_2_O_3_ and the results that the exfoliated flakes preserve the relics of boehmite, except for a reduced thickness of 5–6 nm (from the original 55 nm), it can be inferred that the exfoliation is a natural way of thermal treatment.

## Figures and Tables

**Figure 1 nanomaterials-12-02955-f001:**
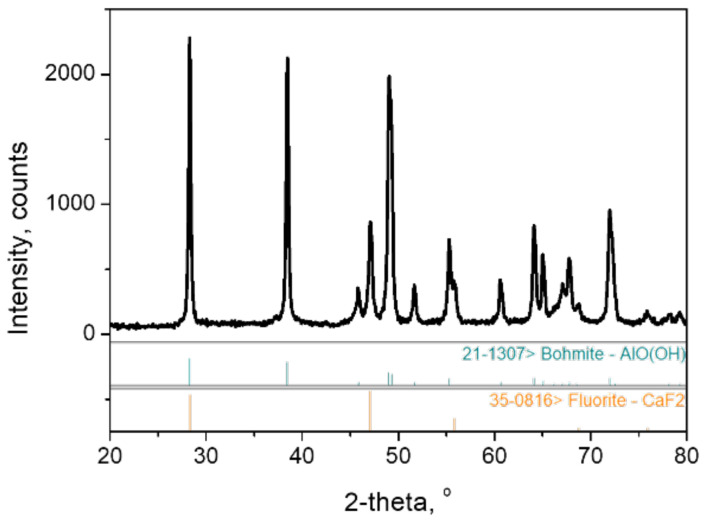
XRD patterns of starting boehmite powders used in this study.

**Figure 2 nanomaterials-12-02955-f002:**
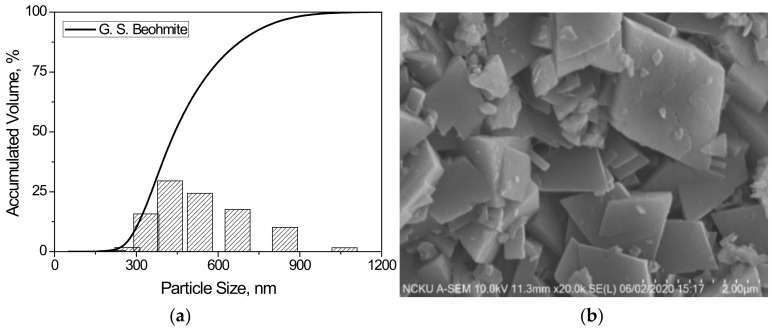
(**a**) Particle size distribution and (**b**) SEM micrograph of starting boehmite powders used in this study.

**Figure 3 nanomaterials-12-02955-f003:**
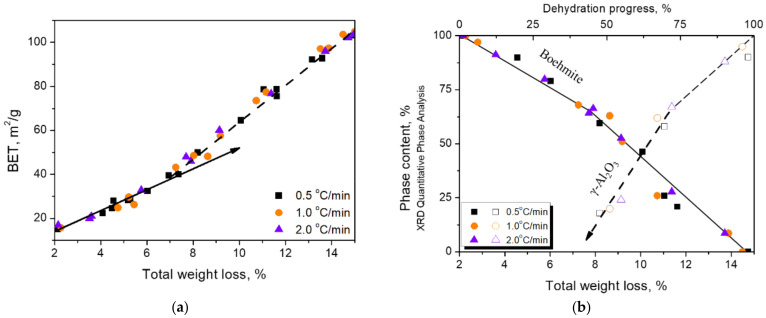
Two stages of BET value increments with the weight loss (**a**). Correlation between the phase contents of boehmite and γ-Al_2_O_3_ and weight loss (**b**).

**Figure 4 nanomaterials-12-02955-f004:**
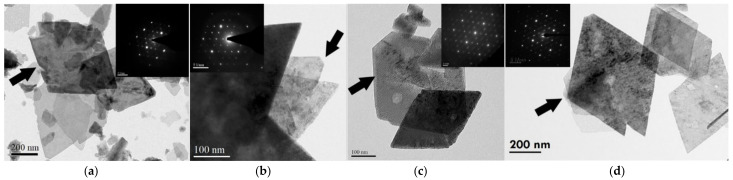
TEM micrographs and diffraction patterns of thermally treated boehmite, (**a**) boehmite particles, (**b**) meta-γ particles, (**c**) γ-Al_2_O_3_ particles pseudomorph after boehmite, and (**d**) γ-Al_2_O_3_ particles.

**Figure 5 nanomaterials-12-02955-f005:**
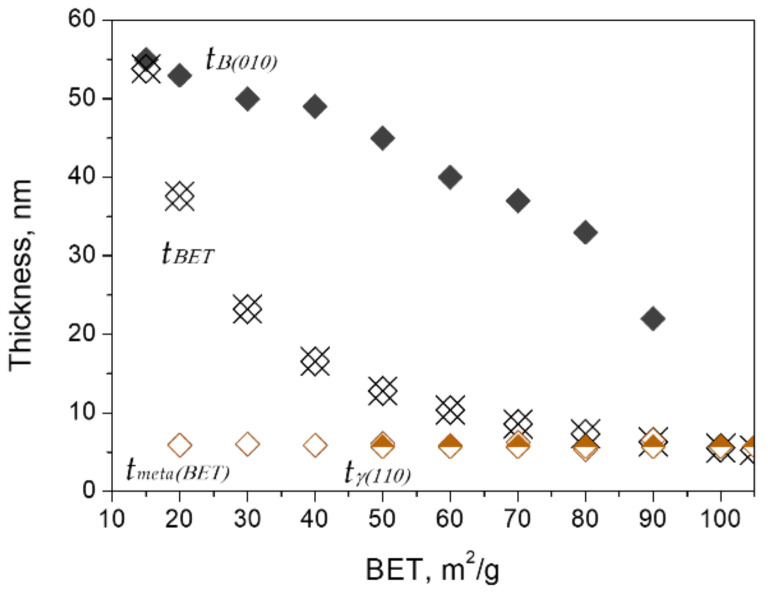
Variations of the average thickness of calcined boehmite samples (*t*_BET_) and flakes of boehmite (*t*_B{010}_), γ-Al_2_O_3_ (*t*_γ{110}_), and meta-γ (*t_meta_*_(BET)_) with exfoliation.

**Figure 6 nanomaterials-12-02955-f006:**
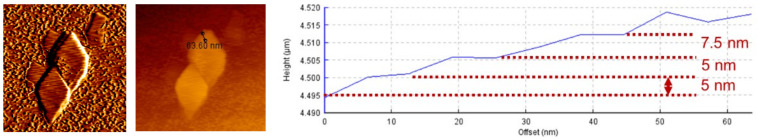
AFM results show the exfoliated flakes displayed approximate thickness of 5 nm.

**Figure 7 nanomaterials-12-02955-f007:**
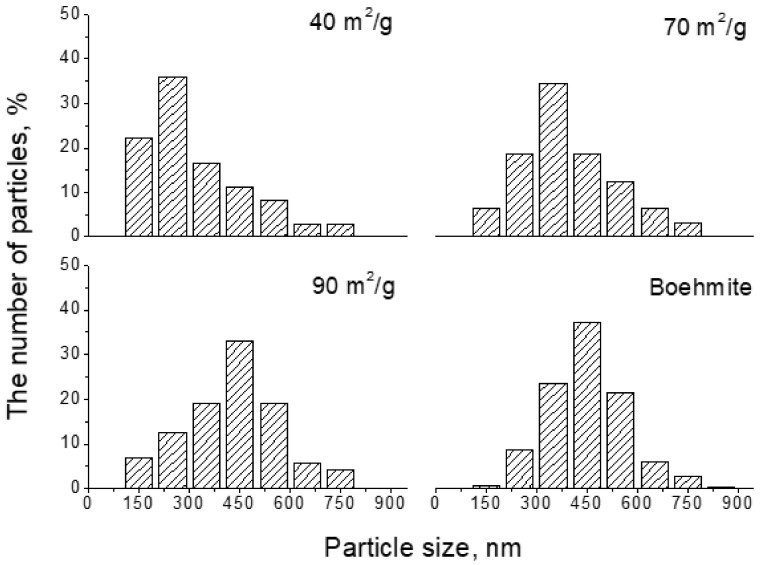
Cross-sectional size distribution of the exfoliated flakes. As the BET value increased, relatively large flakes appeared.

**Figure 8 nanomaterials-12-02955-f008:**
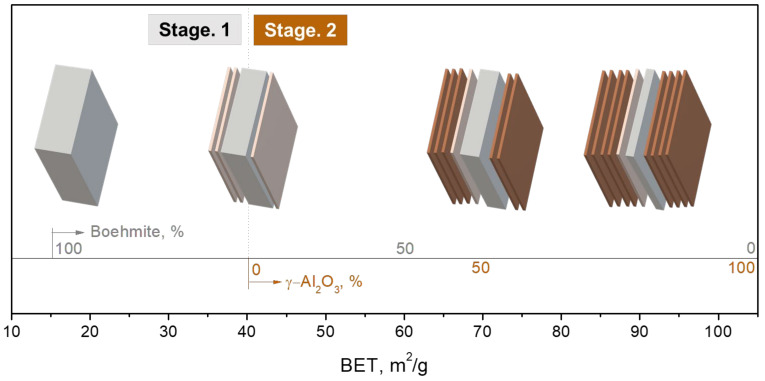
Schematic diagram of the powder exfoliation process at each stage.

**Table 1 nanomaterials-12-02955-t001:** The orientation relationships between boehmite and γ-Al_2_O_3_.

Boehmite		γ-Al_2_O_3_	Relative Difference
a	3.700	→	½ c_γ_	3.962	+7.1%
b	12.227		2 d110	11.2	−8.4%
c	2.868	→	½ d11¯0	2.802	−2.3%
{010}_b_	_ 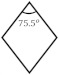 _		{110}_γ_	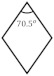	{010}_b_/{110}_γ_	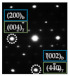

**Table 2 nanomaterials-12-02955-t002:** Basic properties and characteristics of the starting boehmite powder.

Specific surface area	m^2^/g	15
D_50_	nm	450
L.O.I. (loss on ignition)	%	15 + 2 (physical adsorbed)

**Table 3 nanomaterials-12-02955-t003:** Property variations of calcined boehmite samples.

BET, m^2^/g	Phase Contents, %	Particle Shape, %/Thickness, (nm)			Pseudomorph (%, est.)
	Boehmite /(Meta-γ)/γ-Al_2_O_3_	Boehmite	Meta-γ	γ-Al_2_O_3_	
40	70/(30)/0	100/(49) ^2^	--/(5.9) ^1^	0/(5.8) ^1^	--
70	37/(13)/50	74/(37) ^2^	--/(6.2) ^1^	26/(5.7) ^2^	48
90	12/(5)/83	52/(22) ^2^	--/(6.5) ^1^	48/(5.7) ^2^	42
100	0/(0)/100	35/--	--/--	65/(5.7) ^2^	35

^1^: thickness via Equation (1); ^2^: thickness via Equation (2).

**Table 4 nanomaterials-12-02955-t004:** Thermodynamics data.

	Surface	Surface Area,	Surface Energy,	Energy/(Flake),
		m^2^	mJ/m^2^	mJ
Boehmite	−10	3.18 × 10^−13^	455 ^a,^*^,^**	2.38 × 10^−10^
	−101	3.11 × 10^−14^	4585 ^a,^*–3000 ^a,^**	
γ-Al_2_O_3_	−110	3.18 × 10^−13^	2590 ^b,^*–1540 ^b,^**	5.05 × 10^−10^
	−111	8.06 × 10^−15^	4480 ^b,^*–1970 ^b,^**	

^a^ Reference [59]. ^b^ Reference [60]. * Unrelaxed surface energy in vacuum. ** Relaxed surface energy in vacuum. (Adopted for calculation).

## Data Availability

Not applicable.

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
