# Peer review of "Generating Self-Shaped 2D Aluminum Oxide Nanopowders"

_nanomaterials, 2022, doi:10.3390/nano12172955_

Round 1
Reviewer 1 Report
Review report for nanomaterials-1870014
Manuscript ID: nanomaterials-1870014
Manuscript Title: Generating Self-Shaped 2D Aluminum Oxide Nanopowders
This manuscript explains about the Generating Self-Shaped 2D Aluminum Oxide Nanopowders by the thermal-assisted exfoliation phenomena of boehmite particles under moderate heating rates. It is interesting and is excellent to the readers of the journal. It can be reconsidered after minor revision.
1. Need re-written the abstract because it is mixed of past and present. It should be in one form.
2. Fig. 1 can be divided into two figures-the XRD might be one figure and the rest is another figure to represent nicely.
3. Add more references to comparison your works to others.
4. Need English corrections.
Author Response
Point 1: Need re-written the abstract because it is mixed of past and present. It should be in one form.
Response 1:
We rewrite the abstract. Really, there are some mistakes in tense that occurred.
Point 2: Fig. 1 can be divided into two figures-the XRD might be one figure and the rest is another figure to represent nicely.
Response 2:
Thanks for the suggestion. Figure 1 is divided into two figures.
Point 3: Add more references to comparison your works to others.
Response 3:
References including the related techniques and some similar systems such as Al(OH)3, γ-FeOOH, Co(OH)2, and Cd(OH)2 are cited additionally.
Point 4: Need English corrections.
Response 4:
We have found one professor in The Dept. of Foreign Language to help us improve our English writing.

Reviewer 2 Report
The work of the authors is devoted to the study of the formation of the gamma-Al2O3 structure during the heat treatment of boehmite plates. Despite the many works devoted to similar studies, the approach of the authors is quite original and may be of interest to the scientific community. However, it is necessary to take into account the proposed comments.
1. In the introduction, it is necessary to clearly articulate the novelty of the research and indicate the potential contribution of the authors to the acquisition of new knowledge.
2. In the materials and methods section, it is necessary to indicate how the samples were obtained for plotting dependencies in Figures 2, 4, 6.
3. It is necessary to provide XRD pattern for samples after heat treatment.
4. The proposed model of sample exfoliation is doubtful and not convincing enough. If cleavage of the plates occurs and BET(N2) increases, then there must be a distance of at least 2 nm between the layers for the penetration of the N2 molecule. To be more convincing, the authors should provide N2 adsorption-desorption isotherms and TEM images of nanoplate ends before and after heat treatment.
I also want to note that if we get STEM images for the Fig.3c sample, then the authors will see a picture, as in the attached file, from which it can be seen that splitting has occurred not in layers but in domains.
5. The conclusion can be left in this form when more convincing evidence of the model proposed by the authors is provided.

Author Response
Point 1: In the introduction, it is necessary to clearly articulate the novelty of the research and indicate the potential contribution of the authors to the acquisition of new knowledge.
Response 1:
Thanks for the suggestion.
We add lines of sentences describing the potential contribution and the future application of our research in the introduction.
Point 2: In the materials and methods section, it is necessary to indicate how the samples were obtained for plotting dependencies in Figures 2, 4, 6.
Response 2:
Thanks for the suggestion. We have added descriptions in Chapter 2 to indicate How the testing samples were prepared and how the data plotted in Figures 2, 4, and 6 (now are figures 3, 5, and 7.) were obtained.
Point 3: It is necessary to provide XRD pattern for samples after heat treatment.
Response 3:
We add XRD patterns in Appendix A. A serial phase alteration from boehmite to gamma alumina during thermal treatments can be observed.
Point 4: The proposed model of sample exfoliation is doubtful and not convincing enough. If cleavage of the plates occurs and BET(N2) increases, then there must be a distance of at least 2 nm between the layers for the penetration of the N2 molecule. To be more convincing, the authors should provide N2 adsorption-desorption isotherms and TEM images of nanoplate ends before and after heat treatment.
I also want to note that if we get STEM images for the Fig.3c sample, then the authors will see a picture, as in the attached file, from which it can be seen that splitting has occurred not in layers but in domains.
Response 4:
- The proposed model of sample exfoliation is established on four thickness measurement techniques: BET-N2, XRD-Scheerer equation, AFM images, and TEM examination. The cleavage of the plates occurs and BET(N2) increases by which the basic thickness for exfoliated flakes was calculated. The final results of the flake thickness obtained by BET-N2 were 5-6nm (mean thickness). If there were some cases where a distance of at least 2 nm between the layers occurred resulting in the difficulty in penetration of the N2 molecule, the BET values should be smaller. Then the thickness of the flakes must be thinner than that measured by AFM image and XRD-Scherrer equation. It is not the case.
- In our study, the BET method is employed in the relative pressure range P/P0 from 0.05 – 0.3. There are no N2 adsorption-desorption isotherms within this measuring range.
- We presented the TEM images below, the figure (a) demonstrates that the splitting has occurred within layers. And from the electron micrographs figure (b-1), (c), and (d-1), we can see the damage induced by electron beam, because of long time exposure. This can be somewhat similar to the case you provided. Except for this, we did not find domain splitting phenomena. Actually, the whole flake is of single-crystal.
Response 5: We have added more supplementary instructions in Response 4 as evidence of the model proposed in this study.

Round 2
Reviewer 2 Report
Responses to comments are accepted. I wish the authors success in their scientific work.